# FragBuilder: an efficient Python library to setup quantum chemistry calculations on peptides models

Anders S. Christensen[1], Thomas Hamelryck[2] and Jan H. Jensen[1]

[1] Department of Chemistry, University of Copenhagen, Copenhagen, Denmark
[2] Department of Biology, University of Copenhagen, Copenhagen, Denmark

## ABSTRACT

We present a powerful Python library to quickly and efficiently generate realistic peptide model structures. The library makes it possible to quickly set up quantum mechanical calculations on model peptide structures. It is possible to manually specify a specific conformation of the peptide. Additionally the library also offers sampling of backbone conformations and side chain rotamer conformations from continuous distributions. The generated peptides can then be geometry optimized by the MMFF94 molecular mechanics force field via convenient functions inside the library. Finally, it is possible to output the resulting structures directly to files in a variety of useful formats, such as XYZ or PDB formats, or directly as input files for a quantum chemistry program. FragBuilder is freely available at https://github.com/jensengroup/fragbuilder/ under the terms of the BSD open source license.

## INTRODUCTION

Modeling of chemical properties of proteins is a challenging task in modern computational biochemistry, mainly due to the large number of atoms that need to be treated computationally, compared to the computational speed of modern computers. Although theoretical methods to treat large systems are being developed, it is computationally more feasible to investigate properties of small, representative, protein-like structures, such as peptides. For example, calculations on peptides have been used to parametrize protein-specific molecular mechanics force fields, and models for NMR properties of proteins such as chemical shifts and spin-spin coupling constants (*Mackerell, 2004*; *Vila et al., 2009*; *Case, Scheurer & Brüschweiler, 2000*).

Recently, we have used the presented Python library to carry out calculations on peptides modeling the backbone of a protein in the parametrization of amide proton chemical shifts (*Christensen et al., 2013*). Since this study, we have carried out more than 1.5 million quantum mechanical geometry optimization and NMR shielding calculations on peptides in order to extend our model of protein chemical shifts. Naturally, an efficient and stable method is needed in order to generate such a number of peptide models.

Two recent programs that can generate peptide structures are the Ribosome program (*Srinivasan, 2013*) and the PeptideBuilder library (*Tien et al., 2013*). The Ribosome

Corresponding author
Anders S. Christensen,
andersx@nano.ku.dk

program is written in FORTRAN and thus difficult to extend and therefore not ideal for use in an automated, scripting fashion. The PeptideBuilder library is written in Python and is therefore very attractive for this purpose. Our library which is presented here is very similar to PeptideBuilder, but offers a number of additional features which we found necessary for our purpose. Most importantly, our library includes methods for geometry optimization with a molecular mechanics force field, efficient conformational sampling from continuous probability distributions and lastly output to a variety of output formats or, optionally, directly as input file for a quantum chemistry program. Currently Gaussian 09 (*Frisch et al., 2009*) is supported via specialized classes, and nearly 100 additional file formats are supported through the file writer.

## METHODS

FragBuilder is implemented in Python and is a library that can be imported and used in simple Python scripting style. Python is attractive, since a very large number of scientific libraries are already available in Python, and thus easy to extend and combine with new code. FragBuilder is implemented using the Open Babel library as back-end for handling the molecular structure of the peptide via existing classes and methods (*O'Boyle et al., 2011*). The methods present in FragBuilder thus have access to a multitude of existing chemistry and cheminformatics related library routines which are maintained separately by Open Babel. Especially, the code for manipulating a molecular structure, molecular mechanics and file writers from Open Babel are used in FragBuilder. FragBuilder also comes with the BASILISK library which can sample protein backbone and side chain conformations from a joint probability distribution (*Harder et al., 2010*).

The only dependencies for running FragBuilder are the NumPy mathematics library (*Oliphant, 2006*) and Open Babel with Python bindings. These packages are already available through package managers on virtually every recent Linux distribution, or otherwise freely available and open source.

## FUNCTIONALITY AND USAGE

The functionality to create a peptide is implemented in the `Peptide` class which is imported from the `fragbuilder` module. A typical work flow creates a peptide, defines torsion angles, performs a constrained geometry optimization and finally writes the resulting structure to a file. A chart describing a typical use case is displayed in Fig. 1, and detailed examples of the functionality of FragBuilder are given below.

Furthermore FragBuilder has classes to easily access the BASILISK library, read PDB files and write input files for Gaussian 09. An overview of the available class as well as a brief description of each can be found in Table 1.

### Creating peptides

The structure of a peptide molecule is generated as a Python object by using the `Peptide` class instantiated with the sequence as argument. The `Peptide` class has access to classes for each type of residues which each contain a structure for that residue in XYZ format. Routines from Open Babel are then used to automatically rotate, translate, and connect

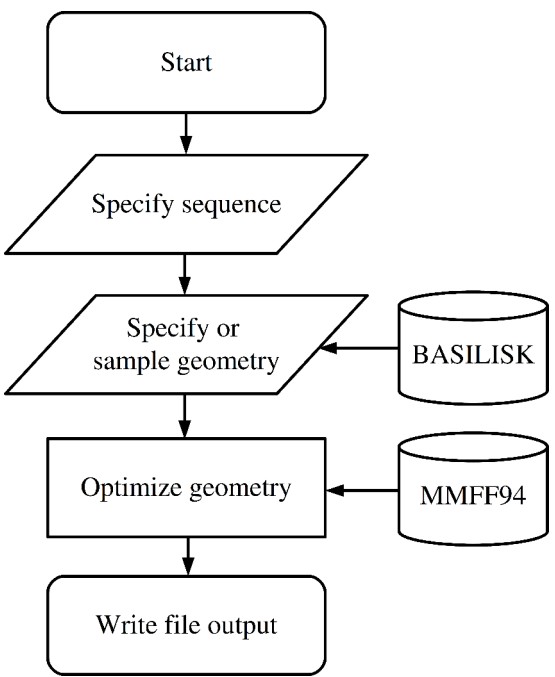

**Figure 1 Flowchart describing the use of FragBuilder.** Simple chart of a common workflow using FragBuilder. First a peptide is generated from the sequence. Then torsion angles are set — either specified manually or sampled through BASILISK and a quick geometry optimization is performed using the MMFF94 force field. Finally, the structure is written to a file.

**Table 1 Overview of classes included in the FragBuilder library.**

| Class name | Description |
|---|---|
| Peptide | Class to create and manipulate a peptide structure and write output files |
| Basilisk_DBN | Wrapper class for direct access to the BASILISK library |
| PDB | Class to extract angles, sequence, etc. from a PDB file |
| G09_opt, G09_NMR, G09_energy | Classes to create input files for QM calculations in Gaussian 09 |

the residues. Finally the structure is stored in the `Peptide.molecule` class variable as an Open Babel `OBMol` object.

The sequence interpreted uses the single letter abbreviation for each amino acid. E.g., `Peptide("GLG")` will create a glycine–leucine–glycine tripeptide molecule which can then be manipulated through the interface. The minimal code to achieve this could be:

```
1 from fragbuilder import Peptide
2 pep = Peptide("GLG")
```

As default values, the $\phi, \psi$ and $\omega$ backbone torsion angles are set to $-120°, 140°$ and $-180°$, which corresponds to a typical extended $\beta$-strand. The side chain torsion $\chi$ angles are set so two neighboring side chains will not have steric clashes when no side chain torsion angle input is given. After the peptide has been instantiated, the structure can

Table 2 **Overview of the basic methods in the** `Peptide` **class.** See the text for detailed descriptions of each method.

| Method name | Description |
| --- | --- |
| `set_bb_angles` | Set the backbone $\phi/\psi$-angles for a residue |
| `set_chi_angles` | Set the side chain $\chi$-angles for a residue |
| `get_bb_angles` | Read the backbone $\phi/\psi$-angles for a residue |
| `get_chi_angles` | Read the side chain $\chi$-angles for a residue |
| `sample_bb_angles` | Sample the backbone $\phi/\psi$-angles for a residue using the BASILISK library |
| `sample_chi_angles` | Sample the side chain $\chi$-angles for a residue using the BASILISK library |
| `optimize` | Perform a molecular mechanics optimization using the MMFF94 force field |
| `regularize` | Perform the regularization procedure to remove steric clashes |
| `write_pdb` | Write the peptide structure to a PDB file |
| `write_xyz` | Write the peptide structure to an XYZ file |
| `write_file` | Write the peptide structure to one of the nearly 100 file types supported by Open Babel |

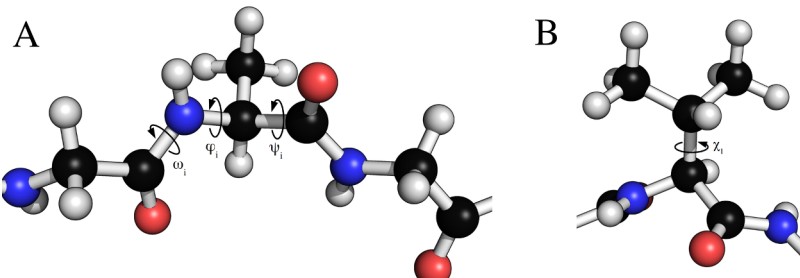

**Figure 2** **Torsion angles that can be treated by FragBuilder.** Examples of dihedral angles that can be set via FragBuilder. In (A) the backbone $\omega, \phi$ and $\psi$ torsion angles are shown for the $i$'th alanine residue of a peptide strand. In (B), the $\chi_1$ torsion angle is shown for a valine side chain.

be manipulated through built-in methods. Several convenient methods of the `Peptide` class are presented in the next sections. An overview of some of the basic methods of the `Peptide` class can be seen in Table 2.

## Setting dihedral angles

The `Peptide` class allows for dihedral angles to be manually specified through setter and getter type functions that set or read backbone and side chain torsion angles. Examples of torsion angles that can be set in FragBuilder are shown in Fig. 2.

For example, making a glycine–leucine–glycine peptide and setting the backbone angles to $\phi = -60.0°$ and $\psi = -30.0°$, and side chain angles to $\chi_1 = 180°$ and $\chi_2 = 60°$ of the leucine (residue 2) can be done through the following code:
```
1 pep = Peptide("GLG")
2 pep.set_bb_angles(2, [-60.0, -30.0])
3 pep.set_chi_angles(2, [180.0, 60.0])
```

This way it is possible to precisely specify dihedral angles manually. This code can be used, for instance, to set up a scan of torsion angles or making peptides with geometries extracted from experimental structures. An example of a scan is shown in Fig. 3. This scan was created in the following manner:

```
1 pep = Peptide("GLG")
2 for i in range(10):
3     pep.set_bb_angles(2, [-120.0, 100.0+20.0*i])
4     pep.write_xyz("pep_%i.xyz" % (i))
```

The method `Peptide.write_xyz()` writes the structure to a file in XYZ format and is described later in this section.

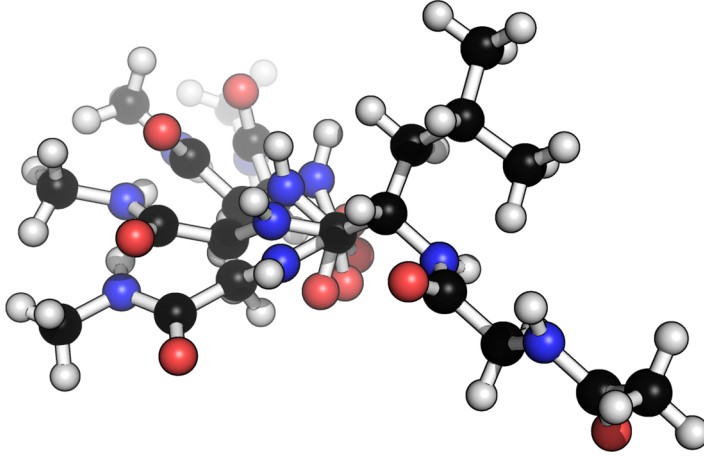

**Figure 3 Example of four different conformers of a glycine–alanine–glycine tri-peptide.** Generated from a scan over the $\psi$ backbone torsion angle of the alanine residue.

## Sampling dihedral angles from BASILISK

In addition to manual specification of torsion angle values, it is possible to set these to values from predefined distributions, such as the Ramachandran-plot for backbone angles or rotamer distributions for side chain angles. This allows for fast and efficient sampling of realistic peptide conformations and rotamer distribution without the need for a molecular dynamics or Monte Carlo simulation. For this purpose FragBuilder includes the BASILISK library and convenient methods to access BASILISK from the `Peptide` class.

BASILISK is a dynamic Bayesian network trained on a large set of representative structures from the Protein Data Bank (*Berman et al., 2000*) and is able to sample backbone angles and side chain angles. BASILISK makes use of directional statistics — the statistics of angles, orientations and directions — to formulate a well-defined joint probability distribution over side and main chain angles. Backbone angles are essentially sampled

from the Ramachandran-plot via BASILISK. Similarly, side chain angles are sampled from corresponding rotamer distributions. The distributions offered by the BASILISK library are continuous, in contrast to most approaches based on discrete rotamer libraries. BASILISK can sample side chain angles either in a backbone conformation-dependent mode or -independent mode (where backbone dependency is the default behavior). The random seed can be set explicitly via the `fragbuilder.set_seed()` function. If no seed is supplied the seeding will be random.

The methods `Peptide.sample_bb_angles()` and `Peptide.sample_bb_angles()` allows the user to simultaneously sample and set the torsion angles of a residue. The methods return the new sets of sampled angles so they are known to the user directly.

The following code will create a glycine–leucine–glycine peptide and set the backbone and side chain angles of the second residue (leucine) to values that are sampled from BASILISK. The values of the sampled angles are stored in the `new_bb` and `new_chi` variables.

```
1 from fragbuilder import Peptide, set_seed
2 set_seed(42)
3 pep = Peptide("GLG")
4 new_bb = pep.sample_bb_angles(2)
5 new_chi = pep.sample_chi_angles(2)
```

It is also possible to get samples from BASILISK via FragBuilder by using the `fragbuilder.Basilisk_DBN` class which provides direct access to the sampler in the BASILISK library. This class is used to obtain samples of $\phi/\psi$ angles from the Ramachandran-plot or sets of $\chi$ angles from rotamer distribution without first creating a peptide.

For instance, a random set of $\chi$ angles (`chi`), $\phi/\psi$ angles (`bb`), and their corresponding log-likelihood (`ll`) in the probability distribution can be obtained as follows (here for a Leucine ("L") residue):

```
1 from fragbuilder import Basilisk_DBN
2 dbn = Basilisk_DBN()
3 # Amino acid type as argument
4 chi, bb, ll = dbn.get_sample("L")
```

10,000 of such samples from the above code was used to create the Ramachandran plot and rotamer distribution of leucine which can be seen in Figs. 5A and 5B, respectively.

## Capping peptides

One aspect of carrying out quantum mechanical calculations on peptide fragments is the way the peptide strands are terminated or capped. This can be important, since the properties calculated from a quantum mechanical calculation may be affected by how the protein is truncated to a model peptide. The specific type of cap is controlled by setting the keywords `nterm` and `cterm` keywords (for the N-terminus and C-terminus, respectively) when the peptide object is created.

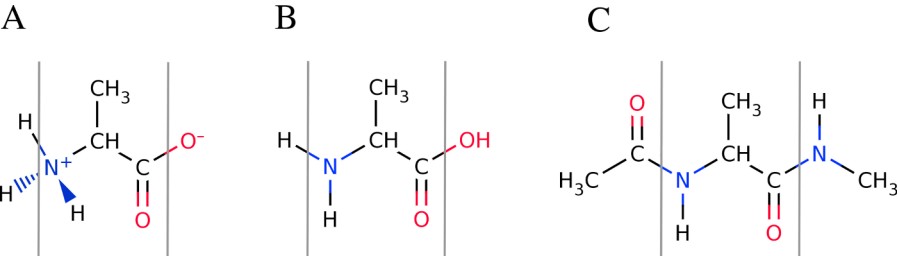

**Figure 4 Overview of the available peptide-capping schemes available in FragBuilder.** All three examples show an alanine residue (shown between a set of gray lines). In (A), the caps are the N- and C-termini in their charged states. In (B) the caps are the N- and C-termini in their neutral states. In (C) the caps are methyl groups. Caps can be mixed and matched according to the user's specifications.

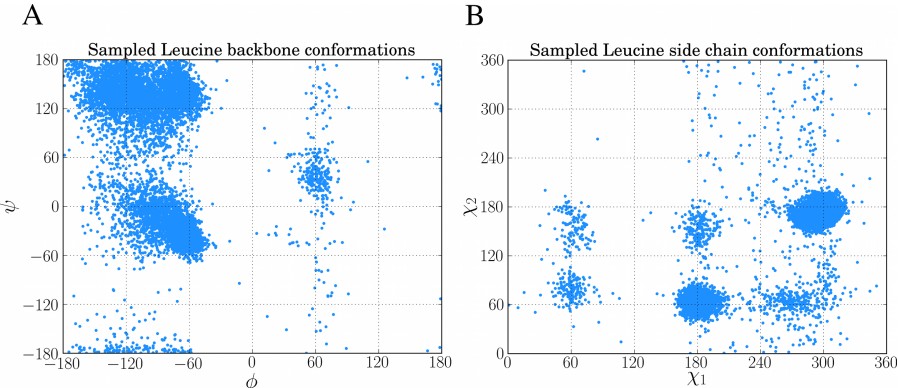

**Figure 5 Examples of sampling dihedral angles through BASILISK in FragBuilder.** 10,000 samples from BASILISK are shown for a leucine residue. $(\phi, \psi)$ backbone torsion angle pairs are shown in (A) and $(\chi_1, \chi_2)$ side chain torsion angle pairs are shown in (B).

By default, FragBuilder generates methyl caps by adding a $CH_3$–$C(=O)$- group to the N-terminus and an -NH–$CH_3$ group to the C-terminus (i.e., if the keywords are not set). This corresponds to setting both keywords (`nterm` and `cterm`) to `"methyl"`. Additionally, it is possible to cap the ends of the peptide as normal N- and C-termini (amine or carboxyl groups, respectively) which can be set to either a charged or a neutral state. A charged or neutral terminus is specified by passing the values `"charged"` or `"neutral"`, respectively. See Fig. 4 for a schematic of the three possible types of caps.

For instance, a glycine–leucine–glycine residue with a positively charged N-terminus and a neutral C-terminus is generated by the following code:

```
1 pep = Peptide("GLG", nterm="charged", cterm="neutral")
```

## Optimization

When generating peptides with a specific set of dihedral angles the structure may, in some cases, contain steric clashes. We found this prevented us from starting quantum mechanical geometry optimization on the structures, even when these were generated to match angles from experimental structures. Typical problems with these structures were

SCF convergence issues and very large molecular gradients which cause the program to fail. In some cases, problems with large molecular gradients may be alleviated by adjusting the step-size in the optimizer, but this must be investigated on a case-to-case basis. It is therefore advantageous to remove steric clashes before any quantum mechanical calculation is carried out.

For the reasons mentioned above, FragBuilder offers specialized molecular mechanics optimization routines, specifically designed to constrain the dihedral angles of peptides while removing steric clashes. Optimization is performed through Open Babel which provides access to several force fields and a number of optimizers. The MMFF94 force field (*Halgren, 1996*) is arguably the most advanced force field for biomolecules in Open Babel and is used exclusively in FragBuilder along with the conjugate gradient method. FragBuilder offers three kinds of optimization methods in the `Peptide` class.

The method `Peptide.optimize()` will perform a conjugate gradient optimization of the peptide with no restraints, until the default convergence criterion of Open Babel is reached ($\Delta E < 1.0 \times 10^{-6}$ kcal/mol or a max of 500 steps). Another option is to impose harmonic constraints on all dihedral angles. This is achieved through an extra keyword, i.e., `Peptide.optimize(constraint=True)`. This will perform a conjugate gradient minimization through Open Babel with harmonic potentials on $\phi$, $\psi$ and $\omega$ backbone angles as well as all side chain $\chi$ angles.

A harmonic potential does not keep torsion angles completely fixed during optimization, and after convergence they deviate slightly from the starting values. For situations where this is problematic, FragBuilder is offering a routine termed "regularizing" which is accessed via the `Peptide.regularize()` method.

Regularizing cycles between a few constrained geometry optimization steps and resetting the dihedral angles to the initially specified angles, until self consistency is reached. A default regularization cycles 10 times between 50 conjugate gradient steps and angle resets. In most cases this converges the constrained optimization to less than $0.002°$ from the specified dihedral angles, which are then set to the specified values.

We found our regularization procedure with flexible bond lengths and angles through the MMFF94 force field to allowing convergence of QM calculations in many cases, which would have been hindered by steric clashes due to fixed bond length and angles.

A similar approach to avoid spurious conformations has been adopted by Vila et al. in the creation of the CheShift chemical shifts predictor, which is parametrized from quantum mechanical calculations on model peptides (*2009*). Here bond angles and lengths are simply set to the standard values of the ECEPP/3 force field (*Nemethy et al., 1992*). Subsequently the internal energy of the peptide is calculated with the ECEPP-05 force field and any conformation with an internal energy >30 kcal/mol is rejected as being unphysical.

Figure 6 shows an example of a tryptophan–aspartate–glycine peptide with methyl caps in which the backbone torsion angles are taken from the experimental structure of xylanase (PDB-code: 1XNB), residues 99–101. This choice of angles causes a clash between a hydrogen bonding O...H pair, and a geometry optimization at the B3LYP/6-31+G(d,p) level in Gaussian 09 could not start (at default settings) due to an excessively large

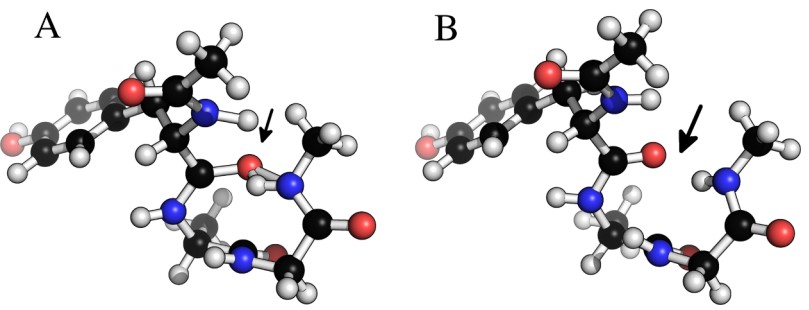

**Figure 6 Removing clashes by regularization in a tryptophan–apartate–glycine peptide.** In (A) the peptide clashes between the amide proton on the C-terminal methyl cap and the amide oxygen in residue 1. In (B) this clash has been removed by constrained relaxation during the regularization procedure. Both structures have identical $\phi, \psi$ and $\omega$ backbone torsion angles.

molecular gradient in the initial geometry. Regularization removes the clash, while retaining the specified dihedral angles, and allows the optimization to proceed.

A peptide can be created and regularized using the following code, which also prints the MMFF94 force field energy in units of kcal/mol:

```
1 pep = Peptide(sequence)
2 # The user can manipulate the structure here
3
4 pep.regularize()
5 print pep.get_energy()
```

## Reading PDB files

While sampling and conformational scanning, etc., are efficient ways to generate new peptide conformations, it can be necessary to extract information about the conformation of a specific protein structure, usually given in PDB format. FragBuilder implements functionality to extract information about the amino acid sequence and dihedral angles from a structure in a PDB formatted file, which can then be stored or passed on in the program, for instance to methods in the `Peptide` class. This is carried out via the `fragbuilder.PDB` class which creates an object from a PDB file and offers methods to read the relevant information.

The following code example illustrates the basic usage of the `fragbuilder.PDB` module, and will print the amino acid type and dihedral angles of residue number 10 in the PDB file `"structure.pdb"`:

```
1 from fragbuilder import PDB
2
3 pdbfile = PDB("structure.pdb")
4 i = 10 # Residue number 10 in this example
5 print pdbfile.get_resname(i)
6 print pdbfile.get_bb_angles(i)
7 print pdbfile.get_chi_angles(i)
```

### File output and interface to QM programs

Open Babel provides very flexible file readers and writers. The `Peptide` class wraps Open Babel with functions to directly write the geometry of a `Peptide` object to a file in XYZ or PDB format. This can be done simply as:

```
1 pep = Peptide(sequence)
2 pep.write_xyz("pep.xyz")
3 pep.write_pdb("pep.pdb")
```

It is also possible to write to any of the nearly 100 formats supported in Open Babel by using the method `Peptide.write_file(filetype, filename)` which offers direct access to Open Babel's `OBConversion.WriteFile()` method. For instance, an input file for the quantum chemistry program GAMESS (*Schmidt et al., 1993*) can be created with the following code:

```
1 pep.write_file("gamin", "pep.inp")
```

Here, the file type argument follows the Open Babel syntax, where `"gamin"` corresponds to the GAMESS input file format.

FragBuilder additionally offers an interface to write input-files for Gaussian 09, beyond the capabilities of Open Babel. Currently, it is possible to set up geometry optimization, single-point energy calculations and calculation of NMR shielding. An example for a simple workflow that will generate a file for geometry optimization of a peptide in Gaussian 09 at the B3LYP/6-31G(d) level (using the `fragbulder.G09_opt` class) is as follows:

```
1 from fragbuilder import Peptide, G09_opt
2
3 pep = Peptide(sequence)
4 # The user can manipulate the structure here
5
6 opt = G09_opt(pep)
7 opt.set_method("B3LYP/6-31G(d)")
8 opt.write_com("pep.com")
```

If no method or basis set is specified, the file writer defaults to PM6 (*Stewart, 2007*) for geometry optimization. Other classes that interface to Gaussian 09 are the `fragbuilder.G09_NMR` and `fragbuilder.G09_energy` classes, which are imported and instantiated similarly.

## CONCLUSION

We have implemented routines to generate peptide models, from either specific geometries or efficient conformational sampling through the BASILISK library. We have furthermore implemented necessary code to perform constrained geometry optimizations of the peptide models, remove steric clashes and prepare the structure for use in a quantum

chemistry program. In addition, the file writers accommodate nearly 100 file formats, and are able to write input files for a number of chemistry programs through an interface to Open Babel.

The `Peptide` class wraps functionality from Open Babel offered through its Python interface. The molecular structure is stored as an Open Babel `openbabel.OBMol` object in the `Peptide.molecule` class variable. This means that developers and users effectively have access to all the tools present in Open Babel to further manipulate the structure, or extend FragBuilder by wrapping and combining functionality from Open Babel.

FragBuilder is open source and published under the BSD 2-Clause license. Note that the packaged BASILISK library is published under the GNU General Public License version 3. FragBuilder is freely available at https://github.com/jensengroup/fragbuilder/ where additional examples and full documentation can be found.

## ACKNOWLEDGEMENTS

The authors would like to thank Casper Steinmann for valuable input during development of FragBuilder.

### Funding

Anders S. Christensen is funded by the Novo Nordisk STAR program. The funders had no role in study design, data collection and analysis, decision to publish, or preparation of the manuscript.

### Grant Disclosures

The following grant information was disclosed by the authors:
Novo Nordisk STAR program.

### Competing Interests

The authors declare there are no competing interests.

### Author Contributions

- Anders S. Christensen conceived and designed the experiments, performed the experiments, analyzed the data, contributed reagents/materials/analysis tools, wrote the paper, prepared figures and/or tables, reviewed drafts of the paper.
- Thomas Hamelryck contributed reagents/materials/analysis tools, wrote the paper, reviewed drafts of the paper.
- Jan H. Jensen conceived and designed the experiments, analyzed the data, wrote the paper, reviewed drafts of the paper.

### Data Deposition

The following information was supplied regarding the deposition of related data:
https://github.com/jensengroup/fragbuilder/.

**Peer**J

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
