# Peer review of "FragBuilder: an efficient Python library to setup quantum chemistry calculations on peptides models"

_PeerJ, doi:10.7717/peerj.277_

## Round 0.1 · original submission · Minor Revisions

As you will see below in the reviewers' comments section, there are some minor concerns that should be carefully addressed before sending an improved version of your manuscript. After submission and verification of the requested changes, we will evaluate the final acceptance of your manuscript.

·

Basic reporting

The paper is rather well written. There are a few cases where the language could be more precise.

The authors say (p. 2) that "calculations on peptides have been used to parametrize ... molecular mechanics force fields, and NMR properties ...". More precisely, this should say "..., and models for the prediction of NMR properties ...".

I am not certain what a "protein-like peptide" is (p. 2). Can the authors clarify what they mean by this expression?

A few language suggestions:
- p. 2: carry out calculationS
- p. 5: FragBuilder offerS

Experimental design

The authors state that typical problems with structures that exhibit sterical clashes are "very large molecular gradients which cause the program to fail" and that this "prevented us from starting quantum mechanical geometry optimizations on the structures" (p. 5). It should be made clear that this is probably not a general problem of QM calculations, but of the particular QM program settings. While performing a force field pre-optimization is generally advantageous, there is no reason why the QM optimizer should not work when the force field optimizer does, even when the gradient is large. (The only exception is when the SCF fails to converge.) Probably the maximum step size in the optimizer simply needs to be adjusted.
Cf. also the statement on p. 6 about a geometry optimization not starting because of a large initial gradient.

Validity of the findings

No Comments

Additional comments

This is useful work and it is laudable that it is being published under open source.

As a suggestion for future work, one could consider adding interfaces to more QM packages, in particular open source ones.

·

Basic reporting

Christensen et al. present a nice python library to generate and manipulate model peptides. The paper is clearly written and it explains the library well. The library itself seems useful; if this library had been available 2 years ago my lab would likely not have written the PeptideBuilder library (Tien et al. 2013, cited in this paper).

I have a few minor comments:

- I would be good to add a table listing all functions and objects provided by fragbuilder, maybe with a 1-sentence description of what they do.

- In Section 3.2.2, the sampling via the BASILISK library could be explained in a little more detail. First, does the code example generate one new configuration, randomly chosen? Would it always be the same if the program is re-run multiple times, or would it differ (i.e., how is the random seed chosen)? Second, could you explain the BasiliskDBN class in a little more detail?

- In Section 3.3, please list all keywords that are available for the nterm and cterm arguments, and state explicitly which keyword is chosen as default. (The way I understand the text, it seems to me that there are three options, "charged", "neutral", and one more that is not named. I may be misunderstanding, but either way this can be presented more clearly.)

- In the first code example on p. 6, I'm not sure the comment is written as meant: "# The user can manipulate the structure here angles here" The phrase "structure here angles here" seems strange.

- In reference Tien et al. 2013, the "B" in "PeptideBuilder" should be capitalized; the word "page" before "1:e80" should be deleted.

- I encourage the authors to review capitalization of all article titles in the references. Remember that bibtex converts everything to lower case. There are several cases where I suspect the capitalization is wrong, e.g. Case et al. 2000.

Experimental design

No Comments.

Validity of the findings

The library installs easily. All provided example programs that I tried ran without error and seemed to produce reasonable results. The generated PDB files look correct.

---

## Round 0.2 · accepted · Accept

After a careful revision of your corrected manuscript, it is a pleasure for me to inform you that we have decided to accept your manuscript for immediate publication.